# Text-Based Person Search in Full Images via Semantic Context Disentangling and Prototype Learning

## Abstract

Text-based Person Search (TBPS) in full images aims to locate a target pedestrian within uncropped images based on natural language descriptions. Existing TBPS methods typically rely on candidate region generation and cross-modal matching. However, in complex scenes,especially those with multiple pedestrians in the image.It is often challenging to distinguish the target pedestrian from the background or other individuals. This leads to limited generalization capabilities. To address these issues, we propose a new TBPS framework named ProtoDis-TBPS, which integrates three key components: Semantic Context Decoupling (SCD), Prototype Embedding Learning (PEL), and a Cross-modal Person Re-identification (ReID) module. Specifically, SCD enhances cross-modal feature discrimination by separating background and irrelevant contextual information. PEL improves the model's robustness in complex scenes by learning prototype features for pedestrian categories. Finally, the ReID module, based on a Transformer architecture, further boosts the accuracy of both text-based pedestrian detection and re-identification in full images. Experiments demonstrate that our proposed method presents a significant challenge to existing approaches in this field.

## 1 Introduction

Text-based person search in full images has significant practical relevance in real-world scenarios, particularly when no images of the target individual are available, and the search relies solely on witness-provided textual descriptions. (Zhang et al., 2024) The goal of this task is to accurately locate a specific pedestrian from complex, uncropped images based on natural language descriptions. However, this task faces numerous challenges, especially when dealing with cluttered backgrounds or multiple pedestrians, where it can be difficult for models to accurately distinguish the target pedestrian. Existing methods often rely on generating a large number of candidate regions for matching, but their robustness and cross-modal matching capabilities are limited, especially in complex scenes.

To address these issues, we propose an end-to-end learning framework based on Transformers, called ProtoDis-TBPS. Inspired by Nies (2023); Zhuang et al. (2023); Gao et al. (2024); Wang et al. (2021); Jiang & Ye (2023),this framework optimizes text-based person search in full images through three key tasks. First, the Semantic Context Decoupling (SCD) task identifies critical information from both visual and textual features, reducing interference from background and irrelevant context. Second, the Prototype Embedding Learning (PEL) task enhances the model's generalization ability in complex scenes by learning prototype representations of pedestrians. Finally, the Cross-modal Person Re-identification (ReID) task performs pedestrian detection and identity recognition based on multimodal features.

By applying this framework to real-world text-based person search tasks in full images, we validated its effectiveness. Experimental results indicate that in complex scenes with multiple pedestrians, the multimodal feature extraction model, which incorporates context

decoupling and prototype learning, can more accurately retrieve the target pedestrian based on textual descriptions.

## 2 RELATED WORKS

In the existing literature, the most relevant work to our proposed method primarily focuses on person search. Approaches in person search are generally based on two commonly used methods from the object detection field: single-stage methods(Liu et al., 2016), where pedestrian detection and feature extraction are performed simultaneously, and two-stage methods(Du et al., 2020), where candidate regions are first generated and then refined through classification and bounding box regression.

Common models for the first approach include the YOLO(Terven et al., 2023) and SSD series(Liu et al., 2016), which are highly efficient and offer an end-to-end training and inference pipeline. However, since detection and feature extraction are done simultaneously, the extracted features may not be detailed enough, potentially affecting retrieval accuracy.

For the second approach, typical models are based on Faster R-CNN, which generates candidate regions through a Region Proposal Network (RPN) and then performs refined detection and classification on each candidate region. Although this method achieves high accuracy and more detailed feature extraction, it often involves computationally intensive modules like Non-Maximum Suppression (NMS)(Bodla et al., 2017).

In addition to these two common approaches, several improved methods have emerged recently, such as anchor-free methods(Zand et al., 2022). These methods no longer rely on predefined anchor boxes but instead detect objects by directly regressing key points or the center of the object, with representative models like FCOS(Tian et al., 2019) and CenterNet(Zhou et al., 2019). Furthermore, there are also Transformer-based methods(Carion et al., 2020), which use various attention mechanisms to perform global feature modeling while simultaneously handling pedestrian detection and feature extraction tasks.

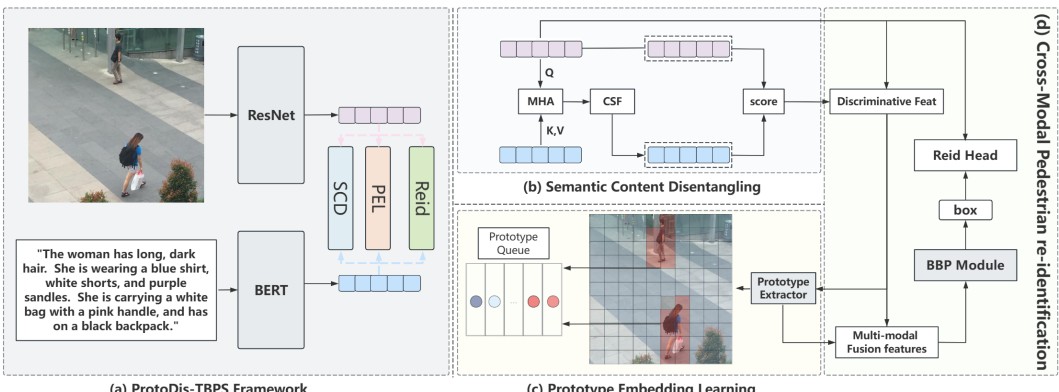

Figure 1: Overview of the ProtoDis-TBPS framework. (a) shows the overall architecture flowchart. The framework consists of three key tasks: (b) Semantic Context Decoupling (SCD) task, which decouples context in the visual and textual modalities separately, enhancing the discriminative power of cross-modal features; (c) Prototype Embedding Learning (PEL) task, which learns prototype representations of pedestrian categories through queue learning, improving the model's robustness in complex scenes; and (d) Cross-modal Person Re-identification (Cross-modal ReID) task, which performs pedestrian re-identification based on multimodal features.

## 3 METHODS

Fig.1 illustrates the overall flow of the ProtoDis-TBPS framework. The primary goal of this framework is to perform person search in full images based on textual descriptions.

Specifically, the framework consists of three key tasks: Semantic Context Decoupling (SCD), Prototype Embedding Learning (PEL), and Cross-modal Person Re-identification (ReID).

First, the image path uses a pre-trained ResNet and Transformer to extract image features, generating a feature representation for the entire image. On the text path, a pre-trained BERT model is used to encode the query description, producing textual feature representations corresponding to the target pedestrian.

In the SCD task, the features from the image and text paths are fed into a Multi-Head Attention (MHA) module, where the image features are used as the query (Q), and the text features serve as the keys (K) and values (V). This process generates a Common Semantic Feature (CSF). The CSF is then processed by a Multi-Layer Perceptron (MLP) and combined with enhanced image features derived from another MLP. These combined features are used to compute a relevance score, thereby fusing salient features from both modalities to generate a discriminative visual representation. This process helps decouple background information and highlight the target pedestrian's features, reducing interference from irrelevant context and background during pedestrian recognition.

In the PEL task, the discriminative visual representation is passed to the Prototype Learning module, which learns typical feature representations of pedestrians. By capturing key features of the target pedestrian and feeding them into a prototype queue for feature-based clustering, this module generates pedestrian features with category-generalized representations. These features are then combined with the discriminative visual representation from the SCD module, forming the Multi-modal Fusion Features.

In the ReID task, the fused multi-modal features and the original image features are fed into a Bounding Box Prediction (BBP) module, which is based on a Transformer and MLP. This module outputs the predicted bounding box of the target pedestrian. Finally, the regressed bounding box and original image features are input into the ReID Head module, where online instance matching is performed to learn and differentiate identity representations of different pedestrians. For more details, please refer to Appendix A.1.

## 4 EXPERIMENTS

## 5 EXPERIMENTS

Finally, we validated the effectiveness of the proposed framework through experiments. We used two public benchmark datasets: CUHK-SYSU-TBPS and PRW-TBPS.(Zhang et al., 2024) The evaluation metrics were the same as those used in traditional TBPS methods, including top-k accuracy (k=1, 5, 10) and mean Average Precision (mAP). For more details, please refer to Appendix A.2.

Tables 1 and 2 present the experimental results on the two benchmark datasets. The results demonstrate that, compared to other Transformer-based methods, our approach achieves significant performance improvements. Through Semantic Context Decoupling (SCD), we are better able to distinguish pedestrians from the background based on textual descriptions, especially in scenarios with many similar pedestrians or complex backgrounds.

## 6 CONCLUSION

In this paper, we propose a cross-modal text-based person re-identification framework, ProtoDis-TBPS, which incorporates Semantic Context Decoupling (SCD) and Prototype

Table 1: Performance of ProtoDis-TBPS on the CUHK-SYSU-TBPS benchmark dataset

| Methods | mAP(%) | Top-1(%) | Top-5(%) | Top-10(%) |
|---------|--------|----------|----------|-----------|
| Ours | 6.45 | 14.48 | 22.86 | 23.39 |

Table 2: Performance of ProtoDis-TBPS on the PRW-TBPS benchmark dataset

| Methods | mAP(%) | Top-1(%) | Top-5(%) | Top-10(%) |
|---------|--------|----------|----------|-----------|
| Ours | 5.43 | 4.10 | 9.28 | 14.26 |

Embedding Learning (PEL) to address the challenge of text-based person search in full images. Through the SCD module, we effectively reduce confusion between the target pedestrian and background or irrelevant context, enhancing the discriminative capability of cross-modal features. Meanwhile, the PEL module successfully captures and inherits the typical features of pedestrians, improving the model's generalization ability in complex scenarios.

Experimental results show that the proposed method significantly outperforms state-of-the-art approaches on multiple benchmark datasets, particularly demonstrating superior robustness and accuracy in handling complex backgrounds and multi-pedestrian scenes.

For future work, we aim to further optimize the cross-modal person re-identification module to better adapt to larger-scale and more complex scenarios. Additionally, we will explore incorporating external knowledge, such as scene context and camera viewpoints, to address real-world challenges like pedestrian occlusion. This will further enhance the model's accuracy and improve its applicability in real-world settings.

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

# A APPENDIX

## A.1 METHOD DETAILS

### A.1.1 TRAINING FRAMEWORK

The ProtoDis-TBPS framework consists of three key subtasks: **Semantic Context Decoupling (SCD)**, **Prototype Embedding Learning (PEL)**, and **Cross-modal Text-based Person Re-identification (ReID)**.

The **SCD** process calculates the similarity between salient objects and contextual information through a cross-modal attention mechanism, enhancing target features while suppressing background interference. Simultaneously, a phrase attention mechanism is employed to enhance contextual information in the description (such as attributes and relationships), further improving the distinction between the target and context in the language representation.

The **PEL** process captures key features of the target pedestrian and performs feature-based clustering by feeding them into a prototype queue. This allows the model to learn and inherit prototype features, ensuring strong robustness and accuracy in varying environments and scene conditions.

The **ReID** process applies cross-modal features and enhanced visual features for cross-modal pedestrian re-identification.

For each image, we use a pre-trained ResNet as the backbone to capture local fine-grained image features. A Transformer is then used as the salient feature embedding module to extract global image features and model the visual context. The image is processed by both modules to obtain a global image feature representation $G_i$.

For the target pedestrian's textual description in each image, we use a pre-trained BERT model(Devlin et al., 2019) to encode the query description, obtaining a global textual feature

representation $G_t$ corresponding to the target pedestrian in the full image. Thus, we have the following representations:

$$G_i = \text{VE}(I), \quad G_t = \text{TE}(T) \tag{1}$$

In the first **Semantic Context Decoupling (SCD) task**, a multi-head attention mechanism (MHA) is used to perform cross-modal information interaction between the global image features $G_i$ and the global textual features $G_t$ associated with the target pedestrian. This results in a fused feature representation $F_{CSF}$, capturing the common semantics between the visual and textual modalities. Next, we pass this fused feature $F_{CSF}$ through a fully connected network (MLP) to further refine and enhance the semantic representation.

To further improve the discriminative power of cross-modal features, we compute the correlation between the visual features $G_i$ and the fused semantic features $F_{CSF}$. Specifically, two independent MLPs are used to project $G_i$ and $F_{CSF}$, respectively, obtaining their correlation score to measure the similarity between the visual and semantic features. Finally, the visual features $G_i$ are adjusted using the correlation score to generate a discriminative visual feature representation $F_{disc}$:

$$F_{CSF} = \text{MHA}(G_i, G_t) \tag{2}$$

$$\text{score} = \text{MLP}_v(G_i) \otimes \text{MLP}_s(F_{CSF}) \tag{3}$$

$$F_{disc} = G_i \odot \text{score} \tag{4}$$

where $\otimes$ denotes the correlation calculation between features, and $\odot$ represents applying the correlation score to the visual features. In this way, the model can focus more on the salient target objects in the image and suppress irrelevant contextual information, thus improving the target distinction capability in complex scenes.

In the second task, **Prototype Embedding Learning (PEL)**, the goal is to capture the key features of pedestrians and generate discriminative multimodal representations. Specifically, the visual features $F_{disc}$ are processed by the **PrototypeExtractor** to extract the prototype feature representation $F_{proto}$ of the pedestrian. These features represent high-level semantic information and are dynamically stored and updated in a **prototype memory queue**.

During training, pedestrian features are clustered to form a discrete set of prototypes. Each prototype represents the cluster center of a pedestrian feature class and is stored in the prototype memory queue $Q_{proto}$ for subsequent inference. During inference, the model searches the prototype queue to find prototypes that semantically match the input features. Specifically, the input visual feature $F_v$ is fused with the pedestrian prototype feature $F_{proto}$ from the prototype memory to generate the final multimodal feature representation $F_{multi}$:

$$F_{proto} = \text{PE}(F_{disc})$$
$$Q_{proto} = \{P_1, P_2, \ldots, P_k\}$$
$$F_{multi} = F_v \oplus F_{proto}$$

Here, the visual features $F_{disc}$ are processed by the PrototypeExtractor (PE) to generate the prototype feature representation $F_{proto}$, which captures high-level semantic information. The prototype memory queue $Q_{proto}$ contains $k$ prototypes (with a default value of $k = 2048$), where each prototype $P_i$ represents a cluster center of a pedestrian feature class, and these prototypes are dynamically updated during training. The input visual feature $F_v$ is concatenated with the prototype feature $F_{proto}$ from the memory queue using the concatenation operation $\oplus$, resulting in the final multimodal feature representation $F_{multi}$.

In the third task, **Cross-modal Person Re-identification (ReID)**, the multimodal feature $F_{multi}$ is combined with the original image features $G_i$ and fed into a **Bounding Box Prediction Module (BBoxPredictor)** based on a **Transformer** and **MLP**. This module generates the bounding box prediction of the target pedestrian.

Next, the regressed bounding box and original image features $G_i$ are passed to the **ReID Head** module, where an **Online Instance Matching (OIM)** loss is used to learn and

differentiate pedestrian identities,commonly used in object detection tasks(Li et al., 2020). During this process, we use **Smooth L1 loss** to measure the error between the predicted and ground truth bounding boxes, yielding the bounding box regression loss $\text{Loss}_{bbox\_reg}$. Additionally, **Binary Cross-Entropy (BCE) loss** is used to measure the difference between the predicted and true labels, yielding the bounding box classification loss $\text{Loss}_{bbox\_cls}$.

We then perform contrastive learning (ITC), commonly used in cross-modal tasks(Li et al., 2022), between the image features $G_{box}$ within the bounding box and the textual features, aligning the visual and textual features at the local level and further reducing the "modality gap" between the two modalities. Finally, the cross-modal person re-identification loss $\text{Loss}_{reid}$ is computed using the OIM loss.

The overall loss function consists of four parts, which measure the errors in the bounding box regression, bounding box classification, and re-identification tasks. The final total loss is expressed as:

$$\text{BBox}_{pred} = \text{BBP}(F_{multi}) \tag{5}$$

$$\text{Loss}_{bbox\_reg} = \text{SmoothL1}(\text{BBox}_{gt}, \text{BBox}_{pred}) \tag{6}$$

$$\text{Loss}_{bbox\_cls} = \text{BCE}(\text{pred\_scores}, \text{gt\_labels}) \tag{7}$$

$$\text{Loss}_{reid} = \text{OIM}(\text{BBox}_{pred}, G_i) \tag{8}$$

$$\text{Loss}_{cma} = L_{ITC}(G_{box}, G_t) \tag{9}$$

$$\text{Loss}_{total} = \alpha \cdot \text{Loss}_{bbox\_reg} + \beta \cdot \text{Loss}_{bbox\_cls} + \gamma \cdot \text{Loss}_{reid} + \theta \cdot \text{Loss}_{cma} \tag{10}$$

where $\alpha, \beta, \gamma, \theta$ are hyperparameters set to 0.2, 0.2, 0.3, and 0.3, respectively. Finally, the loss is jointly optimized throughout the training process.

### A.1.2 INFERENCE

During inference, the query text is passed through the **BERT** model to extract the query text features. Then, multimodal features are computed separately using the query text and gallery images. The cosine similarity between the multimodal features and the query text features is calculated, and a threshold function is applied to reduce the complexity of this process.

The filtered multimodal features are then combined with the visual features of the corresponding gallery images to generate enhanced visual features of the potential target pedestrian. These enhanced visual features are once again compared with the query text features by calculating cosine similarity, which is used for the subsequent ranking and accuracy assessment of the candidate gallery images.

### A.2 EXPERIMENTAL RESULTS AND ANALYSIS

### A.2.1 DATASETS AND EVALUATION METRICS

**Datasets:** We conducted experiments on two recently introduced text-based pedestrian retrieval datasets from full images: **CUHK-SYSU-TBPS** and **PRW-TBPS**. For **CUHK-SYSU-TBPS**, the training set contains 11,206 scene images and 15,080 person bounding boxes with 5,532 different IDs, while the query set includes 2,900 target persons with anchor boxes. Each box in the training set is associated with two textual descriptions, whereas each box in the query set has one description. As for **PRW-TBPS**, the training set contains 5,704 images and 14,897 boxes with 483 different IDs, and the query set contains 2,056 boxes. Each box in the training set has one textual description, while each box in the query set has two descriptions. In our experiments, all dataset splits follow the official training and testing set divisions to ensure comparability of results.

**Evaluation Metrics: Top-K Accuracy (K=1,5,10):** Top-1 accuracy represents the proportion of cases where the first returned candidate matches the textual description correctly. For each text query, the model ranks all candidate pedestrians based on the matching degree between the image features and the text features, returning the bounding boxes of the candidates. The evaluation function computes the cosine similarity between the enhanced visual feature vectors of the candidate pedestrians and the text feature vectors of

the queried person to rank all candidates. Additionally, the IoU (Intersection over Union) threshold between the predicted and ground-truth bounding boxes is dynamically adjusted based on the size of the target pedestrian. Similarly, Top-5 and Top-10 accuracies are computed, reflecting the proportion of correct matches in the top 5 or top 10 candidates.

**mAP (Mean Average Precision):** For each text query, the evaluation function calculates the predicted scores of all candidate pedestrians. The candidates are ranked based on cosine similarity, and the average precision (AP) is computed for each query. The AP is calculated by averaging the precision at different recall levels, and the final mAP value is the average of all queries' APs. The mAP calculation in the evaluation function also considers the IoU between bounding boxes with dynamic thresholds. A higher mAP indicates more stable and robust retrieval performance across different queries and thresholds.

### A.2.2 IMPLEMENTATION DETAILS

**Data Preprocessing:** During the training phase, images are padded and randomly shifted to a fixed size of (640, 640) to enhance the model's robustness to different scenes. Additionally, each image is accompanied by a binary mask, with the same size as the image. The mask marks the actual image region, while the padded parts are set to zero. This mask is used throughout the training process to indicate the valid regions of the image.

**Training Details:** We initialize our visual backbone with a ResNet-50 model, pre-trained on ImageNet for global feature extraction, and a pretrained DETR model for local feature extraction. ResNet-50 enhances visual features by integrating text features, while DETR is utilized to compute semantic similarity with text inputs. For text feature extraction, we employ a **BERT** model. Our model was implemented using PyTorch and tested on an NVIDIA RTX 4090 GPU. The batch size was set to 2, with each image being resized to $640 \times 640$ pixels. Optimization was performed using stochastic gradient descent (SGD) with an initial learning rate of 0.003, which was adjusted during training. The momentum and weight decay for SGD were set at 0.9 and $5 \times 10^{-4}$, respectively. Additionally, we configured a loop queue size of 5000/500 for the Open-set Identity Management (OIM).

