# OpenReview forum: "Text-Based Person Search in Full Images via Semantic Context Disentangling and Prototype Learning"
_ICLR.cc/2025/Conference — Submitted to ICLR 2025_

### Official Review · Reviewer_APKc · 2024-10-27

**Soundness:** 1
**Presentation:** 2
**Contribution:** 1
**Rating:** 1
**Confidence:** 5

**Summary:**

This paper presents a novel framework called ProtoDis-TBPS, which integrates three core components: Semantic Context Decoupling (SCD), Prototype Embedding Learning (PEL), and a Cross-modal Person Re-identification (ReID) module. Specifically, SCD enhances cross-modal feature discrimination by separating background and irrelevant contextual information, while PEL learns prototype features for pedestrian categories to aid the inference process. Moreover, the ReID module supervises the prediction of bounding boxes and IDs, as well as the extraction of pedestrian features.

**Strengths:**

The paper is well-structured, with clear and coherent logic, and appropriate use of methodologies. It innovatively proposes three modules that progressively address the challenges of TBPS, starting from separating background and irrelevant information, to integrating features and prototype extraction, and finally to supervising the prediction results.

**Weaknesses:**

1. The paper falls short of the required page count, with the main text comprising less than 4 pages, while the submission guidelines require 6-10 pages.
2. The proposed method is not compared with existing approaches. Current state-of-the-art (SOTA) performance on the PRW-TBPS and CUHK-SYSU-TBPS datasets stands at 22.17% mAP and 36.78% Rank-1, and 59.62% mAP and 55.35% Rank-1, respectively (MACA: Memory-aided Coarse-to-fine Alignment for Text-based Person Search). In contrast, the proposed method achieves only 5.43% mAP and 4.10% Rank-1 on PRW-TBPS, and 6.45% mAP and 14.48% Rank-1 on CUHK-SYSU-TBPS, which is significantly lower than the SOTA performance.
3. The paper lacks clarity in several key details. For instance, the specific structures of PrototypeExtractor (PE) and Bounding Box Prediction Module (BBP) are not clearly described, and the formulas for the various loss functions are also missing.
4. The paper lacks ablation studies for its components and does not include experiments analyzing the impact of hyperparameters.

**Questions:**

Why does this paper fail to meet the mandatory page count requirement, and why is the performance in the experimental section so poor, with low results and no comparison to existing methods?

---

### Official Review · Reviewer_avYA · 2024-11-01

**Soundness:** 2
**Presentation:** 2
**Contribution:** 2
**Rating:** 3
**Confidence:** 4

**Summary:**

The paper presents a novel framework named ProtoDis-TBPS for locating a target pedestrian within uncropped images based on natural language descriptions. The framework addresses the challenges of distinguishing the target pedestrian from the background or other individuals in complex scenes, which is a common issue in Text-based Person Search (TBPS). ProtoDis-TBPS integrates three key components: Semantic Context Decoupling (SCD), Prototype Embedding Learning (PEL), and a Cross-modal Person Re-identification (ReID) module. SCD enhances feature discrimination by separating background and irrelevant information, PEL improves model robustness by learning prototype features for pedestrian categories, and the ReID module, based on a Transformer architecture, boosts the accuracy of text-based pedestrian detection and re-identification.

**Strengths:**

1. The paper introduces a new end-to-end learning framework for TBPS in full images.
2. The proposed method specifically targets the challenge of distinguishing the target pedestrian in scenes with multiple pedestrians and complex backgrounds, which is a significant advancement over existing methods.

**Weaknesses:**

1. The introduction to each part is so concise that it is hard not to doubt the author's attitude.
2. The introduction of each module of the proposed method is too brief. It should be clear how to do it, why to do it, and why it is effective. The appendix in the current version is unnecessary and should be integrated into the main text.
3. The experimental section mentions comparison with other methods, where are the comparison results?
4. Insufficient experiments. There is a lack of ablation studies to demonstrate the effect of each module.

**Questions:**

Please see Weaknesses

---

### Official Review · Reviewer_m4pg · 2024-11-04

**Soundness:** 1
**Presentation:** 1
**Contribution:** 1
**Rating:** 3
**Confidence:** 4

**Summary:**

This work improves the text-based person search (TBPS) with three modifications. Firstly, a Semantic Context Decoupling (SCD) module for interaction between the image and text, a Prototype Embedding Learning (PEL) for prototype-based metric learning, and a Cross-modal Person Re-identification (ReID) module for the task. Experimental results only show the final performance of the whole network.

**Strengths:**

I feel sorry but from my perspective, this is an unfinished work. Even with all of the contributions, there are no experiments to demonstrate they make sense in this topic. So, I can not find something interesting in this paper.

**Weaknesses:**

This work is far from satisfying, especially in the following points:

1. This paper lacks a detailed introduction to the proposed method. The introduction of the proposed modules is so rough that we can get limited information about how it works. Especially the Prototype Embedding Learning (PEL).

2. This paper lacks detailed experiments to demonstrate all these proposed components make sense. There is no ablation study, no comparison, and no discussion in this paper.

3. This paper's performance is far from satisfying. From the only two tables, the performance of this work seems much lower than the recent works~[1].

[1] Zhang, Shizhou, et al. "Text-based person search in full images via semantic-driven proposal generation." Proceedings of the 4th International Workshop on Human-centric Multimedia Analysis. 2023.

**Questions:**

Please kindly refer to the weakness. In summary, this work at least should add a detailed introduction of each component. How do they work? Detailed experiments to show whether the proposed components make sense also should be added to this paper.

---

> ### Comment · Reviewer_m4pg · 2024-11-26
>
> The author does not provide any response to all of the questions. Therefore, I will keep my rating.

---

### Official Review · Reviewer_BRuh · 2024-11-05

**Soundness:** 2
**Presentation:** 2
**Contribution:** 1
**Rating:** 3
**Confidence:** 4

**Summary:**

This paper focuses on tackling the problem of cluttered backgrounds/multiple pedestrians in TBPS, which have been considered by a few works. To solve this problem, the authors propose an end-to-end learning framework named ProtoDis-TBPS, which integrates three key components: Semantic Context Decoupling (SCD), Prototype Embedding Learning (PEL), and a Cross-modal Person Re-identification (ReID) module.
However, the solution lacks innovation and has limited experiments.

**Strengths:**

The topic is about how to deal with cluttered backgrounds or multiple pedestrians in TBPS, which is worth studying.

**Weaknesses:**

1. The proposed method lacks novelty, which involves cross-attention and prototype learning. There is no technical innovation.
2. Insufficient experiments. Lack of baseline comparison and ablation studies.
3. References are not comprehensive and logical.
4. English writing is not professional enough and drawings are not vivid enough.

**Questions:**

How to understand Table 1 and Table 2?

---

### Meta-Review · Area_Chair_ACzM · 2024-12-20

**Metareview:**

The paper received four negative ratings, with all reviewers inclined to reject it. The paper presents a method involving cross-attention and prototype learning, but key details of the proposed modules, such as Prototype Embedding Learning (PEL), are introduced too briefly, making it difficult to fully understand their functionality and effectiveness. The experimental section is underdeveloped, with insufficient baseline comparisons and no ablation studies to validate the proposed components. The method's performance is significantly lower than recent state-of-the-art works. The paper also fails to provide sufficient details about the structures of critical components, nor does it include necessary loss function formulas. Furthermore, there is no analysis of the impact of hyper-parameters or a discussion on the method's effectiveness. The references are neither comprehensive nor logically connected, and the writing lacks professionalism. The figures are unclear and fail to provide sufficient information. Additionally, the paper does not meet the required page count, which further detracts from its overall quality. Given these issues and the lack of response from the authors, the Area Chair recommends rejecting the paper.

**Additional Comments On Reviewer Discussion:**

The paper received four negative ratings, with all reviewers inclined to reject it.

---

### Decision · Program_Chairs · 2025-01-22

Reject